# *Ascophyllum nodosum* and Silicon-Based Biostimulants Differentially Affect the Physiology and Growth of Watermelon Transplants under Abiotic Stress Factors: The Case of Salinity

**DOI:** 10.3390/plants12030433

**Published:** 2023-01-17

**Authors:** Filippos Bantis, Athanasios Koukounaras

**Affiliations:** Department of Horticulture, Aristotle University, 54124 Thessaloniki, Greece

**Keywords:** NaCl, photosynthetic apparatus, OJIP transient, seaweed, root system architecture, grafted seedlings, *Citrullus lanatus*

## Abstract

Salinization of cultivated soils is a global phenomenon mainly caused by agricultural practices and deteriorates plant production. Biostimulants are products which can be applied exogenously to enhance the plants’ defense mechanism and improve their developmental characteristics, also under abiotic stresses. We studied the potential of two biostimulants, *Ascophyllum nodosum* (Asc) seaweed and a silicon-based (Si), to alleviate the saline conditions endured by watermelon transplants. Three salinity (0 mM, 50 mM, and 100 mM NaCl) treatments were applied in watermelon seedlings transplanted in pots, while the two biostimulants were sprayed in the foliar in the beginning of the experiment. Relative water content was improved by Asc in the high salinity level. The plant area, leaf number, and shoot dry weight deteriorated in relation to the salinity level. However, the root system (total root length and surface area) was enhanced by 50 mM salt, as well as Asc in some cases. The OJIP transient of the photosynthetic apparatus was also evaluated. Some OJIP parameters diminished in the high salinity level after Asc application. It is concluded that after salt stress Asc provoked a positive phenotypic response, while Si did not alleviate the salinity stress of transplanted watermelon.

## 1. Introduction

The salinization of cultivated soils is a global phenomenon that is mainly caused by agricultural practices, such as irrigation with brackish, draining, or even polluted water, combined with a dry climate, resulting in the accumulation of salts in the soil. Saline soils are often rich in Na^+^ and Cl^−^ ions which cause the degradation of soil properties and a reduction in plant growth through toxicity or ionic imbalance [1]. Seedlings have been reported as more sensitive to salt stress compared to mature plants, ultimately showing limitations in their growth. For example, Cuartero and Munoz [2] reported a decrease in the shoot dry weight and root biomass of tomato plants when exposed to salinity conditions due to several reasons, such as lower water and nutrient uptake due to high osmotic pressure and cell limitation or cell death due to ion toxicity. In another study, tomato seedlings in a nursery exhibited decreased above- and below-ground growth, and mineral (Ca, Mg, P, and trace elements) concentration when exposed to saline water conditions [3]. Moreover, salinity has considerable effects on the photosynthetic efficiency and chlorophyll content of plants. For example, tomato and bean exhibited reduced photosynthetic rate due to damaged chloroplast structures from salt stress [4]. Unfavorable environmental and soil conditions, such as increased salinity, are responsible for reduced crop productivity, ultimately threatening worldwide food security [5]. Plant biostimulants have been proposed as a means to mitigate the negative impacts of climate change and potentially achieve yield stability and even enhancement. These novel products trigger molecular and physiological responses when applied to crops, the mechanisms of which are yet to be uncovered [6].

Plant biostimulants are a rather new addition in the list of available products which can be applied exogenous to enhance the plants’ defense mechanism and improve their developmental characteristics [7]. According to the EU Regulation 2019/1009 [8], biostimulants are fertilizing products stimulating plant nutrition with the aim to provide a number of benefits during plant production, such as tolerance to abiotic stress factors and quality trait improvement, among others. Biostimulants can be grouped as non-microbial, including seaweed extracts and silicon shown in the present manuscript, and microbial categories [6,9]. Seaweed extracts, such as *Ascophyllum nodosum*, are an important organic non-microbial biostimulant category showing root-promoting traits. For example, seaweed extracts were shown to enhance the root system development of maize [10]. In addition, silicon has been reported to affect plant–water relations under abiotic stresses including salinity [11], to enhance reactive oxygen (ROS) species scavenging activity [12], to improve photosynthesis [13], and to balance mineral uptake and mobility [14], among other activities. 

Watermelon (*Citrullus lanatus*) is an important species from an economic point of view which is mainly cultivated in eastern Asian and the Mediterranean region [15,16]. The crop is very popular in the Mediterranean basin where cultivated soils typically contain high salinity levels, while climate change conditions are expected to sharply deteriorate in the coming years [17]. Nevertheless, this region still offers the best microclimatic conditions for fruitful watermelon growth; thus, it is imperative that such soils continue to be cultivated, which will further deteriorate the salinity conditions in the upcoming years. Greece is among the highest watermelon producers reaching 12% of the total export value in Europe (FAOSTAT, 2022), with over 60 million euros [18]. The crop is mainly established using seedlings produced in nurseries and afterwards transplanted in the field. Watermelon is a very sturdy crop developing thick leaves and a dense root system. This allows for the plants to withstand any abiotic stress conditions, especially the ones involving water availability, such as drought and salinity, which are mitigated through adjustments in the photosynthetic apparatus [19]. 

The research hypothesis was that the tested biostimulants would positively affect the plant–water relations, the root system development, and possibly the photosynthetic mechanism, especially under the highest salinity level. Our objective was to examine the potential of two biostimulant products, *A. nodosum* seaweed and a silicon-based one, to alleviate the negative effects of salinity in the irrigation water of transplanted watermelon seedlings. To that end, we evaluated the plants’ physiological performance displayed by their chlorophyll fluorescence OJIP transient, as well as determined their overground and root system development. The photosynthetic apparatus can be damaged by environmental stressors. By evaluating the chlorophyll fluorescence OJIP transients which correspond to redox stages within the photosynthetic mechanism, the abovementioned damage can basically be quantified efficiently. 

## 2. Results and Discussion

Relative water content is a measure of the leaves’ ability to withhold water and an essential tool during studies related to inferior water-uptake conditions, such as salinity stress. In the case of watermelon, relative water content was not affected by the different treatments (Table 1). Within salinity treatments, Asc application in treatments involving NaCl and especially at 100 mM showed significantly increased relative water content compared to the other biostimulant treatments (Table 2). Irrigation with saline nutrient solution typically leads to loss of turgor due to osmotic stress [20]. Asc application created the circumstances to alleviate the effects of high salinity. Quite similarly, tomato leaves showed greater relative water content when an *A. nodosum* biostimulant product was applied, and especially when plants were grown under non-saline conditions, which was explained by the activation of osmotic stress tolerance mechanisms, such as proline, fructose, glucose, and sucrose production [21]. In another study, tomato plants grown under saline conditions exhibited higher plant water content when treated with Si which contributed to water dilution and alleviated the toxicity effects of salt [22]. Si deficiency has been associated with greater transpiration since the molecule participates in the cell wall structure of epidermal cells [23]. Moreover, Si application has been reported to form a thin layer over the leaf epidermis leading to reduced transpiration [24]. The above was evident in our case but did not lead to significant responses compared to the other treatments. Nevertheless, it seems that watermelon sturdiness displayed by a dense root system and thick leaves allows for the plants to maintain their water content in high levels leading them to increased photosynthesis and biomass accumulation even under adverse conditions.

Watermelon is a monoecious, cross-pollinated species where male flowers have greater numbers and usually bloom a little earlier compared to female flowers. For the purpose of our study, the experiment finished when the first few flowers bloomed. In our case, male flower number was significantly greater at 0 mM compared to 100 mM NaCl and 100 mM NaCl + Asc (Table 1). Male flowers do not lead to fruits but are indicative of flower initiation which was gradually decelerated by increasing salinity levels. A similar tendency for increased values in 0 mM NaCl treatments was observed in female flowers. However, female flower number did not show significant differences among the different salinity and biostimulant treatments (Table 1). 

Previous studies of our group (e.g., [25]) highlighted the importance of stem diameter as an indicator of seedling quality during the nursery stage. Nevertheless, stem diameter was not significantly affected by the different salinity and biostimulant treatments (Table 1). Obviously, plant development including stem thickening is rapid from transplanting until the flowering stage, which lasted about 20 days in our experiment. During that time, the stem horizontal development was balanced irrespective of salinity level and biostimulant application.

The measurement of plant area covered within a period of time provides a quick and easy evaluation of the plant’s growth and development. In our case, plant area was significantly greater at 0 mM NaCl + Asc than all the 100 mM salinity treatments (Figure 1A). In general, plant area gradually decreased with increasing salinity level regardless of biostimulant application. Moreover, leaf number was significantly greater at all the 0 mM salinity treatments compared to the 50 mM and 100 mM treatments (Figure 1B). Shoot dry weight was significantly greater at 0 mM NaCl and 0 mM NaCl + Asc compared to all the 50 mM and 100 mM treatments (Figure 1C). In the above-mentioned parameters (i.e., plant area, leaf area, and shoot dry weight), the 0 mM treatments showed greater values compared to 50 mM and 100 mM, irrespective of the biostimulants, while 100 mM NaCl showed the lowest values. It is obvious that increased salinity in the nutrient solution caused a severe growth rate deceleration, as also reported in numerous studies including in watermelon experiments (c.f. [26,27]).

The root system is the first organ interacting with the saline nutrient solution, thus among the first to be affected by salinity stress. In our experiment, 0 mM NaCl + Asc showed significantly greater root dry mass than 50 mM NaCl + Si and all the 100 mM treatments (Figure 1D). Additionally, 0 mM and 50 mM showed significantly greater root dry weight compared to 100 mM, irrespective of biostimulants. Within 50 mM and 100 mM salinity levels, Asc showed significantly greater root dry weight values compared to its Si counterpart (Table 2). Furthermore, root surface area and total root length were significantly enhanced at the 50 mM salinity treatments compared to the 0 mM and 100 mM treatments, irrespective of the biostimulants. Specifically, root surface area was significantly greater at 50 mM NaCl + Asc compared to 0 mM NaCl + Si and all the 100 mM treatments, while 50 mM NaCl + Asc showed significantly greater total root length than all the 0 mM and 100 mM treatments (Figure 1E,F). It must be mentioned that both parameters were significantly greater in the 50 mM plants compared to the other salinity treatment, regardless of biostimulant applications. In addition, root surface area at 50 mM NaCl was significantly greater after application of Asc compared to Si and non-biostimulant application (Table 2). Total root length at 0 mM salinity was greater at Asc compared to the other treatments, while at 100 mM NaCl it was also greater at Asc compared to Si (Table 2). Similarly to our findings, in a study involving tomato and *Arabidopsis thaliana*, salinity stress led to shorter roots and a more restricted root network compared to unstressed plants, while root length was enhanced by increasing amounts of the *A. nodosum*-derived biostimulant PSI-475 in both species [21]. Watermelon is a very sturdy crop able to withstand abiotic stress conditions, such as salinity, through adjustments in its root system among other modifications. From our root biomass and architecture analysis it is deducted that a relatively mild addition of 50 mM NaCl in the nutrient solution during the first few weeks after transplantation was beneficial for the development of an extensive root system acting as a positive stress factor. The results are even more profound after application of *A. nodosum* which, in some cases of salinity and biostimulant combinations, enhanced the root system development to a greater extend compared to Si or non-biostimulant treatment, as shown in Table 2. Similar results were reported in a study with maize, *A. nodosum* extracts considerably enhanced root characteristics, such as root length, surface area, diameter, tip number, and fine root length, compared to the non-treated control plants [10].

The photosynthetic apparatus is among the first plant mechanisms that deteriorate in response to several stress factors including salinity. By determining the electron kinetics within the electron transport chain through the OJIP transient evaluation, one can gain rapid and precise information about the effects of a given stressor. In our study we determined the performance index on an absorption basis (PI_ABS_), the probability that an electron moves further than Q_A_ (ψ_E0_), and the maximum quantum yield for primary photochemistry (φ_P0_) which is also addressed as Fv/Fm. We also evaluated the Q_A_ reducing reaction centers per PSII antenna (RC/ABS) corresponding to the active reaction centers on an absorption basis, which is calculated using PI_ABS_, φ_P0_, and ψ_E0_, the relative fluorescence increase between the intersystem carriers and electron end acceptors of PSI (ΔV_IP_), and the relative fluorescence at the J-step (V_J_) (c.f. [28]). 

All OJIP transient parameters maintained typically high levels even under the highest salinity level, proving watermelon sturdiness in abiotic stressors. This is in contrast to the literature, where chlorophyll fluorescence transients are typically reported to decrease with salinity stress [29,30]. PI_abs_ was significantly greater at the 50 mM NaCl + Si, 50 mM NaCl + Asc, and 100 mM NaCl compared to 100 mM NaCl + Asc (Figure 2A and Figure 3B). φ_P0_ was greater at 50 mM NaCl, 50 mM NaCl + Si, and 100 mM NaCl than 100 mM NaCl + Asc (Figure 2B), while ψ_Ε0_ was greater at 100 mM NaCl than 100 mM NaCl + Asc (Figure 2C). Moreover, RC/ABS was greater at all 50 mM treatments compared to 100 mM salinity treatments, irrespective of the biostimulant. Specifically, 50 mM NaCl + Si showed greater values than 0 mM NaCl, 100 mM NaCl + Si, and 100 mM NaCl + Asc (Figure 2D and Figure 3A). V_J_ was greater at 100 mM NaCl + Asc compared to 100 mM NaCl (Figure 2E). In addition, at the high salinity level, 100 mM NaCl, Asc led to greater V_J_ but lower PI_abs_ and ψ_E0_ values compared to the non-biostimulant counterpart (Figure 3C and Table 2). ΔV_IP_ was greater at all 100 mM treatments compared to the 0 mM salinity treatments, irrespective of the biostimulant used. Specifically, 100 mM NaCl showed greater values than 0 mM NaCl, 0 mM NaCl + Asc, and 50 mM NaCl (Figure 2F). It is clear that Asc application at the highest salinity level (100 mM NaCl) slightly diminished the electron transport activity as mostly displayed by PI_ABS_, ψ_E0_, and V_J_. Nevertheless, plants treated with Asc exhibited enhanced root system development in some cases of 50 mM or 100 mM salinity, indicating that salt stress might have promoted a positive phenotypic response, a eustress. On the other hand, Ye et al. [27] reported an improvement in the net photosynthesis, Fv/Fm (i.e., φ_P0_), and chlorophyll content of salt-stressed watermelon seedlings treated with arbuscular mycorrhizal fungi. Regarding Si, it has been reported to ameliorate the inhibition of photosynthetic activities under saline conditions in various horticultural species, such as cucumber [31], tomato [22], and zucchini [13]. Possible mechanisms derived from Si application include the greater accumulation of pigments related to light absorption, as well as the protection of the chloroplast ultrastructure under saline conditions through the conservation of the double membranes and subsequently the granae [32]. However, this was not evident in the case of watermelon where Si-treated plants performed similarly to their non-treated counterparts. In another study, *A. nodosum* also showed no effect of chlorophyll fluorescence parameters of spinach leaves even though gas exchange was significantly affected, pointing to fine water relations which reduced the stomatal closure [33]. The above finding possibly applies to the case of Asc in our study, which maintained the plant growth even after decreased fluorescence parameters.

Among the salinity treatments, relative chlorophyll content was significantly greater at 0 mM NaCl + Si compared to 50 mM NaCl + Asc, 100 mM NaCl, and 100 mM NaCl + Si (Table 1), while in general the 0 mM treatments had significantly greater values compared to the 100 mM treatments, irrespective of biostimulant application. This result demonstrates the negative effect of salinity on the content of photosynthetic pigments. Similarly, Ikuyinminu et al. [21] reported that *A. thaliana* showed lower chlorophyll content under salinity stress, and especially when no *A. nodosum* biostimulant was applied. Moreover, photosynthetic pigments (chlorophylls and carotenoids) were reported to be accumulated when tomato plants were treated with *A. nodosum* seaweed [34].

Elevated salinity levels in the nutrient solution are known to result in ROS overproduction [35]. Salinity-stressed cucumber treated with Si exhibited improved activity of superoxide dismutase (SOD), glutathione reductase (GR), guaiacol peroxidase (GRX), ascorbate peroxidase (APX), and dehydroascorbate reductase (DHAR), while H_2_O_2_ and thiobarbituric acid reactive substances (TBARS) were reduced, indicating that Si mitigated the oxidative damage caused by salt stress [31]. Moreover, salt-stressed *Brassica napus* plants showed increased activities of catalase and cell wall peroxidase when treated with Si [36]. A study with *A. thaliana* grown under salt stress and treated with copper chlorophyllin biostimulant showed increased plant growth, demonstrated by greater shoot length and fresh shoot weight when the product was applied [37]. The authors also found upregulated 5 NADPH/respiratory burst oxidases, genes involved in ROS signaling, in plants treated with copper chlorophyllin, indicating the biostimulant’s effect in ROS scavenging activity. Respiratory burst oxidases have been reported to be highly expressed in salt-tolerant barley mutants [38].

## 3. Materials and Methods

### 3.1. Plant Material and Cultivation

The watermelon seedlings were produced and provided by a commercial nursery (Agris S.A., Kleidi, Imathia, Greece) in May 2021. On the following day, the experiment was initiated in a plastic greenhouse (N 40.536; E 22.995). Watermelon “Celine F1” was used as scion, and *Cucurbita maxima* × *C. moschata* “TZ-148” hybrid was used as rootstock, forming a vigorous grafted watermelon seedling. Upon receiving the seedlings, they were transplanted in 1 L plastic pots. The substrate was peat and perlite at a ratio of 2:1 and the pots were irrigated at full water capacity. Afterwards, the transplants were moved onto a bench inside the plastic greenhouse. During cultivation, day/night temperature was about 18/32 °C, and relative humidity was maintained at 65 ± 10%. Photoperiod was 15 h with maximum light intensity of 1185 μmol m^−2^ s^−1^.

### 3.2. Salinity and Biostimulant Treatments

Beginning at two days after transplanting (DAT) and every two days onwards the transplants were watered with 200 mL Hoagland [39] nutrient solution (100% strength; pH 6.5; EC 2.6 mS cm^−1^) but with different NaCl concentrations. Three levels of salinity were applied at 15 pots in each case; 0 mM, 50 mM, and 100 mM NaCl, which were determined through a preliminary experiment. The plants were grown until blooming of the first few flowers at DAT 20.

At the same day, only DAT 2, two biostimulant products were sprayed in the foliar: a silicon-based (Si) biostimulant, which was applied at 30 kg/ha, and *A. nodosum* (Asc) seaweed biostimulant, which was applied at 4 L/ha. Si consists of >85% SiO_2_ (*w/v*). Asc consists of *A. nodosum* extract 19.5% (*w/v*), K_2_O (13.7%), and P_2_O_5_ (9.8%). Specifically, each biostimulant was sprayed at five plants per salinity treatment and 15 plants in total. Afterwards, five pots/replicates were randomized on the bench in a randomized complete block design (RCBD) (Figure 4). Salinity level of 0 mM and non-biostimulant application were considered as the Control treatment.

### 3.3. Measurements amd Analyses

The plants grew until the first flowers bloomed in DAT 20. At the beginning of flowering, we determined the plant area through images analyzed with software (WinRHIZO Pro software, Regent Instruments Inc., Quebec, QC, Canada), the relative chlorophyll content (CCM-200, Opti-Sciences, Hudson, NH, USA), the number of leaves, and the number of male and female flowers. Moreover, we determined the shoot dry weight, as well as leaf relative water content as follows: RWC [%] = [(FW − DW)/(TW − DW)] × 100. FW: fresh weight; TW: turgor weight; DW: dry weight. Root parameters, such as dry weight, surface area, and total length, were also determined. Shoots and roots were oven-dried for 3 days at 72 °C to obtain the dry weight values.

Analysis of the chlorophyll fluorescence OJIP transient was conducted using a chlorophyll fluorometer (Pocket PEA, Hansatech, King’s Lynn, UK), according to Strasser et al. [28]. Specifically, measurements were conducted on leaves adapted to the dark for 20 min. Chl a fluorescence was induced by 1 s pulses of 650 nm red light at a light intensity of 3500 μmol m^−2^ s^−1^. In the present manuscript, we present important quantum parameters of the electron transport chain, such as PI_ABS_, φP0, ψE0, RC/ABS, V_J_, and ΔV_IP,_ which are described and discussed in the previous section.

Statistical analysis was conducted with specialized software (SPSS 23.0, IBM Corp., Armonk, NY, USA). ANOVA and Tukey post hoc analyses were conducted at significance level α = 0.05.

## 4. Conclusions

Transplanted watermelon seedlings were subjected to three salinity levels (0 mM, 50 mM, and 100 mM NaCl) and treated with two biostimulants, *A. nodosum* seaweed and a silicon-based one, to test their potential to alleviate the salt-stress responses. Obviously, shoot growth parameters, such as plant area, leaf number, and shoot dry weight, as well as the male flower number, were negatively affected depending on the salinity level. The first barrier between saline water and plant, the root system displayed by the total root length and surface area, as well as the relative water content were enhanced by Asc in some cases of salinity. Silicon exhibited a negative effect on root system development but did not show positive or negative effects in other morphophysiological parameters. Additionally, one of our aims was to study the electron transport chain using the OJIP transient under a combination of salinity stress and biostimulant application. Regarding the OJIP transient evaluation, some parameters (i.e., PI_ABS_, ψ_E0_, and V_J_) showed lower values after Asc application in the high salinity level, indicating diminished photosynthetic activity due to decreased electron transfer flow from the plastoquinone pool to the PSI reaction center. However, some of the same watermelon plants also revealed greater root system development showing that elevated salinity promoted a positive phenotypic response. In conclusion, salt stress led to declined watermelon growth and development, but *A. nodosum* application attenuated the negative impacts at high salinity levels. Finally, it should also be mentioned that watermelon transplants are very sturdy; thus, their performance was mostly diminished at the highest salinity level, while the effect of NaCl or the biostimulants was not considerable compared to what was expected before the start of the experiment.

## Figures and Tables

**Figure 1 plants-12-00433-f001:**
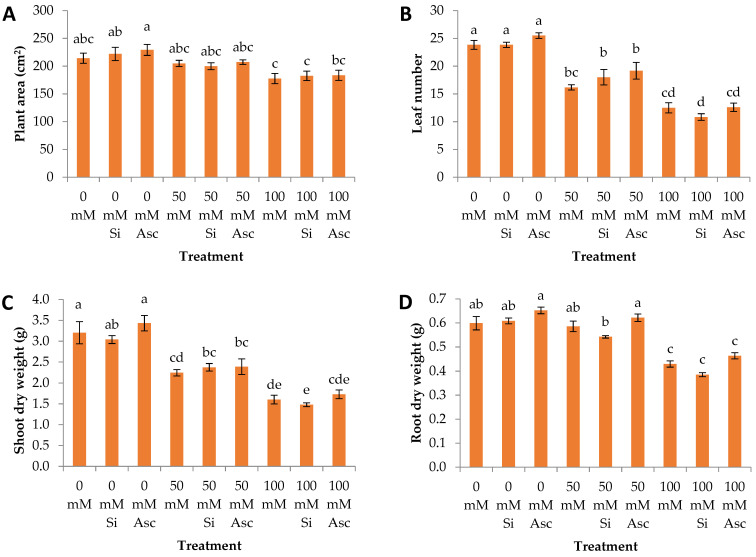
(**A**) Plant area, (**B**) leaf number, (**C**) shoot dry weight, (**D**) root dry weight, (**E**) root surface area, and (**F**) total root length of watermelon transplants irrigated with nutrient solution of different NaCl concentration and sprayed with two biostimulants. Within a bar, average values (n = 5) followed by different letters are significantly different (a < 0.05).

**Figure 2 plants-12-00433-f002:**
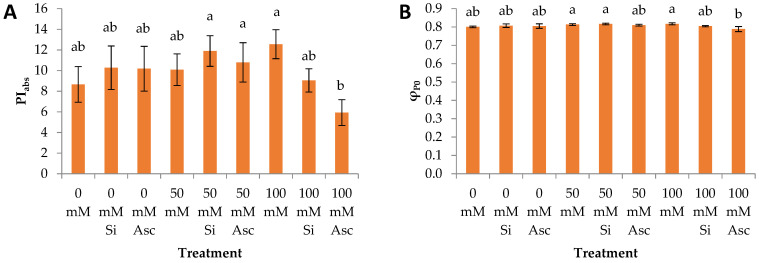
(**A**) PI_abs_, (**B**) φ_P0_, (**C**) ψ_E0_, (**D**) RC/ABS, (**E**) V_J_, and (**F**) ΔV_IP_ of watermelon transplants irrigated with nutrient solution of different NaCl concentration and sprayed with two biostimulants. Within a bar, average values (n = 5) followed by different letters are significantly different (a < 0.05).

**Figure 3 plants-12-00433-f003:**
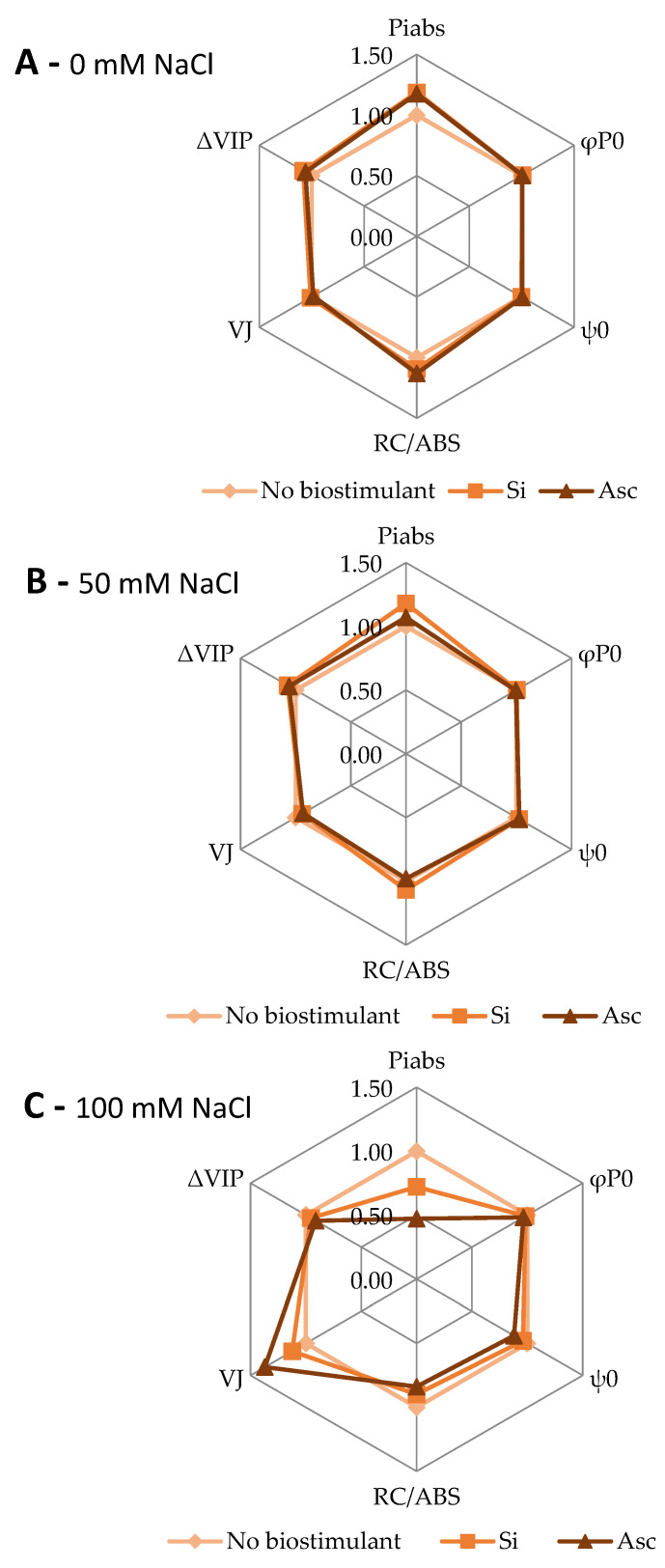
OJIP parameters normalized to the values of the non-biostimulant treatments. (**A**) 0 mM NaCl, (**B**) 50 mM NaCl, and (**C**) 100 mM NaCl. Post hoc statistical analysis is displayed in Table 2.

**Figure 4 plants-12-00433-f004:**
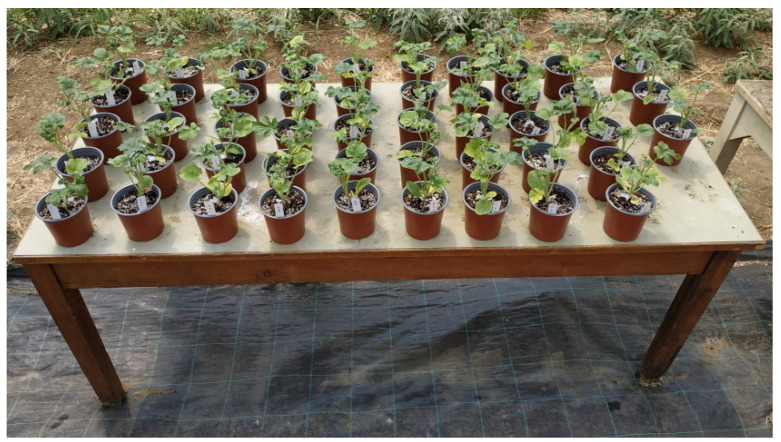
Watermelon seedlings transplanted in pots, irrigated with nutrient solution of different NaCl concentration, and sprayed with two biostimulants.

**Table 1 plants-12-00433-t001:** Floral and physiological parameters of watermelon transplants irrigated with nutrient solution of different NaCl concentration and sprayed with two biostimulants. Within a column, average values (n = 5) followed by different letters are significantly different (a < 0.05). RWC: relative water content.

Treatments	RWC %	Male Flower Number	Female Flower Number	Male/Female Flower %	Stem Diameter (mm)	Rel. Chlorophyll Content
0 mM NaCl	73.8 ± 2.7 ^a^	2.33 ± 0.66 ^a^	0.67 ± 0.40 ^a^	78/22	6.98 ± 0.44 ^a^	81.7 ± 8.1 ^ab^
0 mM NaCl + Si	73.3 ± 2.7 ^a^	2.17 ± 0.82 ^ab^	0.00 ± 0.00 ^a^	100/0	6.75 ± 0.26 ^a^	104.3 ± 10.4 ^a^
0 mM NaCl + Asc	71.3 ± 0.7 ^a^	0.17 ± 0.16 ^ab^	0.33 ± 0.26 ^a^	34/66	6.41 ± 0.15 ^a^	78.1 ± 11.2 ^ab^
50 mM NaCl	75.3 ± 1.7 ^a^	1.00 ± 0.52 ^ab^	0.17 ± 0.21 ^a^	85/15	6.42 ± 0.09 ^a^	65.1 ± 15.00 ^ab^
50 mM NaCl + Si	74.8 ± 1.6 ^a^	1.50 ± 0.67 ^ab^	0.00 ± 0.17 ^a^	100/0	6.90 ± 0.18 ^a^	71.7 ± 9.5 ^ab^
50 mM NaCl + Asc	77.5 ± 1.7 ^a^	1.00 ± 0.52 ^ab^	0.00 ± 0.00 ^a^	100/0	6.71 ± 0.18 ^a^	47.7 ± 7.2 ^b^
100 mM NaCl	72.2 ± 0.7 ^a^	0.00 ± 0.00 ^b^	0.00 ± 0.00 ^a^	0/0	6.68 ± 0.16 ^a^	52.3 ± 10.3 ^b^
100 mM NaCl + Si	72.7 ± 0.9 ^a^	0.17 ± 0.17 ^ab^	0.00 ± 0.00 ^a^	100/0	7.00 ± 0.17 ^a^	48.5 ± 4.9 ^b^
100 mM NaCl + Asc	77.9 ± 1.2 ^a^	0.00 ± 0.00 ^b^	0.00 ± 0.00 ^a^	0/0	6.49 ± 0.21 ^a^	68.5 ± 4.8 ^ab^

**Table 2 plants-12-00433-t002:** Tukey post hoc statistical analysis of plant biostimulant (PB) applications within the different salinity levels. Within a row and a salinity level, different letters denote significant differences (a < 0.05). RWC: relative water content.

Parameter	0 mM NaCl			50 mM NaCl			100 mM NaCl		
	No PB	Si	Asc	No PB	Si	Asc	No PB	Si	Asc
RWC	a	a	a	a	a	a	b	b	a
Male flowers	a	a	a	a	a	a	a	a	a
Female flowers	a	a	a	a	a	a	a	a	a
Stem diameter	a	a	a	a	a	a	a	a	a
Rel. chl. content	a	a	a	a	a	a	a	a	a
Plant area	a	a	a	a	a	a	a	a	a
Leaf number	a	a	a	a	a	a	a	a	a
Shoot dry weight	a	a	a	a	a	a	a	a	a
Root dry weight	a	a	a	ab	b	a	ab	b	a
Root surface area	a	a	a	b	c	a	a	a	a
Total root length	b	b	a	a	a	a	ab	b	a
V_J_	a	a	a	a	a	a	b	ab	a
PI_abs_	a	a	a	a	a	a	a	ab	b
φ_P0_	a	a	a	a	a	a	a	a	a
ψ_E0_	a	a	a	a	a	a	a	ab	b
RC/ABS	a	a	a	a	a	a	a	a	a
ΔV_IP_	a	a	a	a	a	a	a	a	a

## Data Availability

Data sharing is not applicable to this article.

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
