# Peer review of "Ascophyllum nodosum and Silicon-Based Biostimulants Differentially Affect the Physiology and Growth of Watermelon Transplants under Abiotic Stress Factors: The Case of Salinity"

_plants, 2023, doi:10.3390/plants12030433_

Round 1

Reviewer 1 Report

The manuscript entitled " Ascophyllum nodosum and silicon-based biostimulants differentially affect the physiology and growth of watermelon transplants under abiotic stress factors: the case of salinity " is an interesting article to read on and indeed will be beneficial for the researchers working in the area of abiotic stress. However, I have some suggestions and comments for the overall improvement of this article-

1.      The title needs to be rewritten – why do authors wish to include abiotic stresses and the case of salinity both--, please directly convey the massage with salinity or salt stress, instead above said. In my point of view the last lines of the title much reflect a title for a review article-  please rewrite it in a more reflective manner –

2.      Please provide some information about OJIP in the introduction section for a better understanding of readers and relating with the MS.

3.      Material and methods section- Please make it clearer when the experiment was terminated after which date of transplantation, although it has been mentioned in the results section it will be better to include it in M&M too. Secondly how the roots were dried, what temperatures, oven, or shade dry etc. include all details about each measurement and analysis conducted.

4.      Results and discussion-  Table 1- data regarding male and female flowers count, I guess the pattern of representation would have been much better if the data was presented in % percent form, instead of average reflective value –

5.      I would suggest the inclusion of the figures of the pot experiment conducted by the authors for better visualization and representation of the data. 

Author Response

The manuscript entitled " Ascophyllum nodosum and silicon-based biostimulants differentially affect the physiology and growth of watermelon transplants under abiotic stress factors: the case of salinity " is an interesting article to read on and indeed will be beneficial for the researchers working in the area of abiotic stress. However, I have some suggestions and comments for the overall improvement of this article-

Response: The authors would like to express their gratitude to the reviewers for the time they invested for assessing our manuscript and their useful comments and suggestions. The comments were responded one-by-one below.

  1. The title needs to be rewritten – why do authors wish to include abiotic stresses and the case of salinity both--, please directly convey the massage with salinity or salt stress, instead above said. In my point of view the last lines of the title much reflect a title for a review article- please rewrite it in a more reflective manner –

Response: The manuscript is part of the postdoctoral studies of Dr. Filippos Bantis (first author) which also includes the effect of other abiotic stress factors on watermelon transplants. Our plan was to publish our results as a series of articles with similar title. We recently published our results about drought stress, while this manuscript (salinity) follows if accepted. Therefore, we prefer not to alter our title.

  1. Please provide some information about OJIP in the introduction section for a better understanding of readers and relating with the MS.

Response, L83-86: Information about OJIP where included in the introduction as suggested.

  1. Material and methods section- Please make it clearer when the experiment was terminated after which date of transplantation, although it has been mentioned in the results section it will be better to include it in M&M too. Secondly how the roots were dried, what temperatures, oven, or shade dry etc. include all details about each measurement and analysis conducted.

Response: The termination date (DAT 20) was included in the materials and methods, L298 and L312. The roots were oven-dried for 3 days at 72 C. It is now mentioned in L320.

  1. Results and discussion- Table 1- data regarding male and female flowers count, I guess the pattern of representation would have been much better if the data was presented in % percent form, instead of average reflective value –

Response, L127: In Table 1, a column depicting the floral parameters as a percentage was included as suggested.

  1. I would suggest the inclusion of the figures of the pot experiment conducted by the authors for better visualization and representation of the data.

Response, L310: A figure (figure 4) depicting the pot arrangement was included in the manuscript as suggested.

Reviewer 2 Report

The authors obtained large amount of data concerning the effect of salinity and biostimulants on the growth of watermelon. Their results should be published. Nevertheless, there are several comments to the text.

The authors repeatedly emphasize in the text that “this is the first evaluation of OJIP transient for salt-stressed plants after application of Asc” (see lines 18-19, 73-75, 178-180, 290-293). Such a statement sounds strange and gives an unfavorable impression of the article. Chlorophyll fluorescence and, in particular, JIP-test is extremely widely used in many agricultural tasks. If such a negligible innovation like usage JIP-test for studying the effect of some particular substance is highlighted, this suggests that it is the main and probably the only result of the paper. I am sure that this claim should be removed from the paper if the authors would not like to led the reader to this conclusion.

It seems that the main result of the work is negative. Namely, most of the parameters studied by the authors are insensitive to both salinity and biostimulants. Even where there are statistically significant differences, they are usually very small (except for the number of leaves and flowers, shoot and root weight, and, probably, Chl content). It seems to me that there is no need to be shy about this, and this insensitivity should be formulated as the main conclusion which undoubtedly has its own value.

If the authors would like to highlight that there is a significant effect of some biostimulant in some conditions, it would be useful to show the corresponding data in a separate figure(s). The presentation of all the data in a unified manner (as they are presented at the figures now) is very convenient for presenting the overall picture, but with regard to those few effects that the authors might like to emphasize it causes scrolling blindness.

The paper almost lacks the Discussion. It seems to me that discussion is not only a comparison of the figures obtained in the work with ones obtained by others with other model plants. Discussion should contain interpretation of the data (rather than simple statements that this parameter is higher, and this one is lower): what is the physiological meaning of the obtained data; why some parameters are sensitive to any treatment and some are not; what is the integral picture of physiological changes concluded from the entire data set?  

In particular, if the authors do not want to restrict their conclusions to the statement that all the studied JIP test parameters are practically insensitive to the studied effects, and would like to describe the behavior of parameters in detail, then it is necessary to explain how the authors interpret this behavior. E.g., the statement that some parameter “was significantly greater at the 50 mM Si, 50 mM Asc, and 100 mM compared to 100 mM Asc” is not sufficient; it is necessary to explain (at least to hypothesize) the corresponding changes in plant physiology.

Some minor remarks.

Lines 78-83 — this paragraph should be moved to Introduction.

Table 1 — why letters indicating (non)significant difference are only in two columns?

Fig. 1 C,D — why is fresh weight of shoots but dry weight of roots is given? Could you present both sets of data in a unified manner (both fresh, or both dry, or both fresh and dry)?

Table 2, Lines 194, 234 (Fig. 2D) — what is 10RC/ABS? In Materials and methods section, only RC/ABS is indicated.

Line 257 — what is “Silicon-based (Si) biostimulant”? Please, describe it in more details.

Line 290 — morphopsysiological parameters => morphophysiological

Author Response

The authors obtained large amount of data concerning the effect of salinity and biostimulants on the growth of watermelon. Their results should be published. Nevertheless, there are several comments to the text.

Response: The authors would like to express their gratitude to the reviewers for the time they invested for assessing our manuscript and their useful comments and suggestions. The comments were responded one-by-one below.

The authors repeatedly emphasize in the text that “this is the first evaluation of OJIP transient for salt-stressed plants after application of Asc” (see lines 18-19, 73-75, 178-180, 290-293). Such a statement sounds strange and gives an unfavorable impression of the article. Chlorophyll fluorescence and, in particular, JIP-test is extremely widely used in many agricultural tasks. If such a negligible innovation like usage JIP-test for studying the effect of some particular substance is highlighted, this suggests that it is the main and probably the only result of the paper. I am sure that this claim should be removed from the paper if the authors would not like to led the reader to this conclusion.

Response: Indeed, our aim was not to test only the photosynthetic mechanism, but also morphological parameters and practical aspects of biostimulants used to alleviate salinity stress. Therefore, we removed these statements (L18, L86, L198, and L343) to avoid confusion as suggested.

It seems that the main result of the work is negative. Namely, most of the parameters studied by the authors are insensitive to both salinity and biostimulants. Even where there are statistically significant differences, they are usually very small (except for the number of leaves and flowers, shoot and root weight, and, probably, Chl content). It seems to me that there is no need to be shy about this, and this insensitivity should be formulated as the main conclusion which undoubtedly has its own value.

Response, L351-355: We agree with your comment. A statement was included in the manuscript as suggested.

If the authors would like to highlight that there is a significant effect of some biostimulant in some conditions, it would be useful to show the corresponding data in a separate figure(s). The presentation of all the data in a unified manner (as they are presented at the figures now) is very convenient for presenting the overall picture, but with regard to those few effects that the authors might like to emphasize it causes scrolling blindness.

Response, L276: Thank you for your comment. A figure depicting the OJIP parameters (Figure 3) was included for better visualization of the photosynthetic apparatus results.

The paper almost lacks the Discussion. It seems to me that discussion is not only a comparison of the figures obtained in the work with ones obtained by others with other model plants. Discussion should contain interpretation of the data (rather than simple statements that this parameter is higher, and this one is lower): what is the physiological meaning of the obtained data; why some parameters are sensitive to any treatment and some are not; what is the integral picture of physiological changes concluded from the entire data set? 

Response: The Discussion was enhanced as suggested. Please see L211-213, L233-236, L238-242, and L249-251.

In particular, if the authors do not want to restrict their conclusions to the statement that all the studied JIP test parameters are practically insensitive to the studied effects, and would like to describe the behavior of parameters in detail, then it is necessary to explain how the authors interpret this behavior. E.g., the statement that some parameter “was significantly greater at the 50 mM Si, 50 mM Asc, and 100 mM compared to 100 mM Asc” is not sufficient; it is necessary to explain (at least to hypothesize) the corresponding changes in plant physiology.

Response: Thank you for your comment. As mentioned in a comment above, we agree that many parameters proved less sensitive to our salinity levels as well as biostimulant applications, thus a statement was included in the conclusions (L351).

A figure depicting the OJIP parameters (Figure 4) was included for better visualization of the photosynthetic apparatus results.

Some minor remarks.

Lines 78-83 — this paragraph should be moved to Introduction.

Response: The paragraph was moved to the introduction (L41-47) as suggested.

Table 1 — why letters indicating (non)significant difference are only in two columns?

Response, L127: Letters were only provided when there were significant differences. However, now we included letters even in parameters with non-significant differences.

Fig. 1 C,D — why is fresh weight of shoots but dry weight of roots is given? Could you present both sets of data in a unified manner (both fresh, or both dry, or both fresh and dry)?

Response, L206: Shoot fresh weight was substituted with dry weight for better consistency as suggested (Figure 1C). No differences occurred with the statistical analyzes.

Table 2, Lines 194, 234 (Fig. 2D) — what is 10RC/ABS? In Materials and methods section, only RC/ABS is indicated.

Response, L133: Thank you for the observation. We corrected the term throughout the manuscript. In our case, the correct term is 10RC/ABS. Basically, it is RC/ABS multiplied by 10 for better visualization.

Line 257 — what is “Silicon-based (Si) biostimulant”? Please, describe it in more details.

Response, L302-303: The Si and Asc were described in the materials and methods.

Line 290 — morphopsysiological parameters => morphophysiological

Response, L341: Corrected. Thank you for the observation.

Reviewer 3 Report

The manuscript entitled Ascophyllum nodosum and silicon-based biostimulants differentially affect the physiology and growth of watermelon transplants under abiotic stress factors: the case of salinity" is based on original research experiment and the presented results therein broaden the knowledge of plant sciences. To examine the potential of two biostimulant products, A. nodosum seaweed and a silicon-based, to alleviate the negative effects of salinity in the irrigation water of transplanted watermelon seedlings was the main aim of the work. Authors conducted experiment in controlled conditions, during which chlorophyll fluorescence (JIP – test), chlorophyll content index and morphological features were measured. There is no doubt that this work is in the scope of Plants journal. The publication presents some interesting studies. The work delivers some interesting results and can be important source of valuable information.

The introduction is properly composed. The materials and methods section contains the basic requested elements and provide information about the experimental preparations, analyses. However, details about growth conditions were omitted. The data analysis is generally properly provided. The results show valuable information. The obtained data are discussed sufficiently.

However, the authors made some shortcomings that must be corrected before the publication of the work:

1) Authors did not make a research hypothesis. In fact, it is not known what mechanism of the factors the authors want to explain.

2) Table 1 and 2: parameter units must be specified. Now it is not known, for example, whether flowers are their number or weight.

3) Figure 2 and table 2: I do not know such parameter like 10RC/ABS.

4) MM section: under what conditions was the experiment conducted? The air temperature, humidity, photoperiod and light intensity must be specified.

5)  MM section: although Pocket PEA has only one measurement protocol, but in my opinion it is worth providing its settings (measurement time, intensity of actinic light, etc.)

6) I suggest to add additional radar plot with all JIP-test parameters. Moreover, the charts with OJIP curve are in my opinion very important and would significantly enrich this work. Below are the publications in which the authors can find such charts. These are works on the use of JIP test to analyze the impact of various stress factors, so they can be used in introduction or discussion:

DÄ…browski P., et al. 2021. Photosynthetic efficiency of Microcystis ssp. Under salt stress. Environmental and Experimental Botany 186.

Esmaeilizadeh M., et al. 2021. Manipulation of light spectrum can improve the performance of photosynthetic apparatus of strawberry plants growing under salt and alkalinity stress. PLoS ONE 16(12): e0261585.

DÄ…browski P., et al. 2019. Exploration of chlorophyll a fluorescence and plant gas exchange parameters as indicators of drought tolerance in perennial ryegrass. Sensors 19, 2736.

7) Conclusions: I suggest to pay attention to the issue of which element of the photosynthetic apparatus was protected from stress by biostimulators

I would like to underline that my remarks are auxiliary and not undertake the quality and importance of the paper.

Author Response

The manuscript entitled “Ascophyllum nodosum and silicon-based biostimulants differentially affect the physiology and growth of watermelon transplants under abiotic stress factors: the case of salinity" is based on original research experiment and the presented results therein broaden the knowledge of plant sciences. To examine the potential of two biostimulant products, A. nodosum seaweed and a silicon-based, to alleviate the negative effects of salinity in the irrigation water of transplanted watermelon seedlings was the main aim of the work. Authors conducted experiment in controlled conditions, during which chlorophyll fluorescence (JIP – test), chlorophyll content index and morphological features were measured. There is no doubt that this work is in the scope of Plants journal. The publication presents some interesting studies. The work delivers some interesting results and can be important source of valuable information.

The introduction is properly composed. The materials and methods section contains the basic requested elements and provide information about the experimental preparations, analyses. However, details about growth conditions were omitted. The data analysis is generally properly provided. The results show valuable information. The obtained data are discussed sufficiently.

Response: The authors would like to express their gratitude to the reviewers for the time they invested for assessing our manuscript and their useful comments and suggestions. The comments were responded one-by-one below.

However, the authors made some shortcomings that must be corrected before the publication of the work:

1) Authors did not make a research hypothesis. In fact, it is not known what mechanism of the factors the authors want to explain.

Response, L76-78: The research hypothesis was included in the manuscript.

2) Table 1 and 2: parameter units must be specified. Now it is not known, for example, whether flowers are their number or weight.

Response, L127: Thank you for the observation. Units were included in Table 1, except for relative chlorophyll content which is dimensionless. Table 2 shows the statistical comparison among the treatments for each parameter, thus units were omitted.

3) Figure 2 and table 2: I do not know such parameter like 10RC/ABS.

Response: 10RC/ABS corresponds to the active reaction centres on absorption basis. It is now mentioned in L195.

4) MM section: under what conditions was the experiment conducted? The air temperature, humidity, photoperiod and light intensity must be specified.

Response, L290-292: The climatic information were included in the manuscript as suggested.

5)  MM section: although Pocket PEA has only one measurement protocol, but in my opinion it is worth providing its settings (measurement time, intensity of actinic light, etc.)

Response, L325-326: The PEA setting were included in the manuscript as suggested.

6) I suggest to add additional radar plot with all JIP-test parameters. Moreover, the charts with OJIP curve are in my opinion very important and would significantly enrich this work. Below are the publications in which the authors can find such charts. These are works on the use of JIP test to analyze the impact of various stress factors, so they can be used in introduction or discussion:

DÄ…browski P., et al. 2021. Photosynthetic efficiency of Microcystis ssp. Under salt stress. Environmental and Experimental Botany 186.

Esmaeilizadeh M., et al. 2021. Manipulation of light spectrum can improve the performance of photosynthetic apparatus of strawberry plants growing under salt and alkalinity stress. PLoS ONE 16(12): e0261585.

DÄ…browski P., et al. 2019. Exploration of chlorophyll a fluorescence and plant gas exchange parameters as indicators of drought tolerance in perennial ryegrass. Sensors 19, 2736.

Response, L211-213, L276: Radar plots of OJIP parameters (Figure 3), as well as relevant discussion and citations were included in the manuscript as suggested.

7) Conclusions: I suggest to pay attention to the issue of which element of the photosynthetic apparatus was protected from stress by biostimulators

Response, L346-347: An explanation was included in the conclusions as suggested.

I would like to underline that my remarks are auxiliary and not undertake the quality and importance of the paper.

Round 2

Reviewer 1 Report

The comments and suggestions on the manuscript entitled " Ascophyllum nodosum and silicon-based biostimulants differentially affect the physiology and growth of watermelon transplants under abiotic stress factors: the case of salinity has been worked in well. I appreciate the author's effort and work to improve the overall structure of the manuscript.

I have no further comments or suggestions and recommend the article for further perusal.   

Author Response

The authors would like to express their gratitude to the reviewers for the time they invested for assessing our manuscript and their useful comments and suggestions.

Reviewer 2 Report

In the revised version, the authors have taken into consideration almost all my comments. The paper can be published after a minor revision — there are three comments regarding the new fragments of the text.

Lines 83-85, “By evaluating the chlorophyll fluorescence OJIP transients; the redox stages within the photosynthetic mechanism; the abovementioned damage can basically be quantified efficiently” —

the sentence seems to be not concorded. Probably, the semicolons should be replaced with “and” or numbering. E.g.: “By evaluating the chlorophyll fluorescence OJIP transients and the redox stages within the photosynthetic mechanism, the abovementioned damage can basically be quantified efficiently”; or: “By evaluating i) the chlorophyll fluorescence OJIP transients and ii) the redox stages within the photosynthetic mechanism, the abovementioned damage can basically be quantified efficiently”.

Lines 263-265 (Fig. 3): “100 mM Asc” can be only interpreted as “concentration of Asc equal to 100 mM”. The indication of the added biostimulant and of salt concentration should be separated: “100 mM NaCl, Asc”, “100 mM NaCl + Asc”, or “Asc and 100 mM NaCl”, or “Asc, 100 mM NaCl”; or better to indicate 0, 50 or 100 mM NaCl in a separate line, and to mark the symbols with “no biostimulants”, “Si” and “Asc”. The same applies to the text (Table 1, lines 114, 158, and others).

If the authors prefer to use RC/ABS multiplied by 10 for better visualization, this can be done in the axis title in the plot, but by no means in the definition of the value! When used in the axis title, the 10 should be separated from the name of the value by a multiplication symbol or by an asterisk. For example, see the article by Zhang et al. 2021, Figure 3C, or Rizzo et al. 2014, Figures 3 and 4. Moreover, in the Figure 3, the normalized parameters are shown, hence multiplication of the parameter by any number does not change the normalized value and should be omitted. The same applies to the Table 2 (only letters are given there!).

Thus, the designation RC/ABS x10 could have sense only in the Fig.2D axis title, and should be replaced by RC/ABS in any other places of the text, including Fig.2 legend, Fig.3, Table 2, and “Materials and Methods” section!

Everything else is OK.

References

1. Zhang, C., Zeng, G., Zhang, R., Tang, Y., Liu, Q., & Jiang, T. (2021). Tunable nonlinear optical responses of few-layer graphene through lithium intercalation. Nanophotonics, 10(10), 2661-2669. (https://www.degruyter.com/document/doi/10.1515/nanoph-2021-0173/html?lang=de  )

2. Rizzo, F., Zucchelli, G., Jennings, R., & Santabarbara, S. (2014). Wavelength dependence of the fluorescence emission under conditions of open and closed Photosystem II reaction centres in the green alga Chlorella sorokiniana. Biochimica et Biophysica Acta (BBA)-Bioenergetics, 1837(6), 726-733. (https://www.sciencedirect.com/science/article/pii/S0005272814000565 )

Author Response

In the revised version, the authors have taken into consideration almost all my comments. The paper can be published after a minor revision — there are three comments regarding the new fragments of the text.

Response: The authors would like to express their gratitude to the reviewers for the time they invested for assessing our manuscript and their useful comments and suggestions. The comments were responded one-by-one below.

Lines 83-85, “By evaluating the chlorophyll fluorescence OJIP transients; the redox stages within the photosynthetic mechanism; the abovementioned damage can basically be quantified efficiently” — the sentence seems to be not concorded. Probably, the semicolons should be replaced with “and” or numbering. E.g.: “By evaluating the chlorophyll fluorescence OJIP transients and the redox stages within the photosynthetic mechanism, the abovementioned damage can basically be quantified efficiently”; or: “By evaluating i) the chlorophyll fluorescence OJIP transients and ii) the redox stages within the photosynthetic mechanism, the abovementioned damage can basically be quantified efficiently”.

Response, L83-84. The sentence was altered for better concordance, as suggested.

Lines 263-265 (Fig. 3): “100 mM Asc” can be only interpreted as “concentration of Asc equal to 100 mM”. The indication of the added biostimulant and of salt concentration should be separated: “100 mM NaCl, Asc”, “100 mM NaCl + Asc”, or “Asc and 100 mM NaCl”, or “Asc, 100 mM NaCl”; or better to indicate 0, 50 or 100 mM NaCl in a separate line, and to mark the symbols with “no biostimulants”, “Si” and “Asc”. The same applies to the text (Table 1, lines 114, 158, and others).

Response. We agree with your comment. Treatments in the text are now indicated in a proposed manner; for example “100 mM NaCl + Asc”. In Figure 3, symbols were marked as “no biostimulant”, “Si” and “Asc” to be clearer.

If the authors prefer to use RC/ABS multiplied by 10 for better visualization, this can be done in the axis title in the plot, but by no means in the definition of the value! When used in the axis title, the 10 should be separated from the name of the value by a multiplication symbol or by an asterisk. For example, see the article by Zhang et al. 2021, Figure 3C, or Rizzo et al. 2014, Figures 3 and 4. Moreover, in the Figure 3, the normalized parameters are shown, hence multiplication of the parameter by any number does not change the normalized value and should be omitted. The same applies to the Table 2 (only letters are given there!).

Thus, the designation RC/ABS x10 could have sense only in the Fig.2D axis title, and should be replaced by RC/ABS in any other places of the text, including Fig.2 legend, Fig.3, Table 2, and “Materials and Methods” section!

Response. We agree with your comment. The multiplication was removed from the whole manuscript including the figures (figure 2D was recalculated) and tables.

Everything else is OK.

References

  1. Zhang, C., Zeng, G., Zhang, R., Tang, Y., Liu, Q., & Jiang, T. (2021). Tunable nonlinear optical responses of few-layer graphene through lithium intercalation. Nanophotonics, 10(10), 2661-2669. (https://www.degruyter.com/document/doi/10.1515/nanoph-2021-0173/html?lang=de )

  1. Rizzo, F., Zucchelli, G., Jennings, R., & Santabarbara, S. (2014). Wavelength dependence of the fluorescence emission under conditions of open and closed Photosystem II reaction centres in the green alga Chlorella sorokiniana. Biochimica et Biophysica Acta (BBA)-Bioenergetics, 1837(6), 726-733. (https://www.sciencedirect.com/science/article/pii/S0005272814000565 )

Reviewer 3 Report

Dear Authors,

All my comments have been taken into account. I believe that the work can be published in its current version.

With best regards,

Author Response

(The authors gave the same response as above.)
